# Analysis of Air Pollutant Emissions for Mechanized Rice Cultivation in Korea

Gyu-Gang Han [1], Jun-Hyuk Jeon [2], Yong-Jin Cho [2,3,4], Myoung-Ho Kim [2,3,4] and Seong-Min Kim [1,2,3,4,*]

[1] Department of Agricultural Convergence Technology, Graduate School, Jeonbuk National University, Jeonju 54896, Korea; dt200v@jbnu.ac.kr

[2] Department of Agricultural Machinery Engineering, Graduate School, Jeonbuk National University, Jeonju 54896, Korea; splinter9608@jbnu.ac.kr (J.-H.J.); choyj@jbnu.ac.kr (Y.-J.C.); Myoung59@jbnu.ac.kr (M.-H.K.)

[3] Department of Bioindustrial Machinery Engineering, College of Agriculture and Life Sciences, Jeonbuk National University, Jeonju 54896, Korea

[4] Institute for Agricultural Machinery & ICT Convergence, Jeonbuk National University, Jeonju 54896, Korea

[*] Correspondence: smkim@jbnu.ac.kr; Tel.: +82-63-270-2617

**Abstract:** In Korea, rice is a major staple grain and it is mainly cultivated using various types of agricultural machinery. Air pollutants emitted from agricultural machinery have their origins mainly from the exhaustion of internal combustion engines. In this study, the emission characteristics of five main air pollutants by the European Environment Agency's Tier 1 method for rice cultivation were analyzed. Diesel is a main fuel for agricultural machinery and gasoline is generally used only for rice transplanters as a fuel in Korea. Tractors consume 46% of total fuel consumption and 56% of diesel fuel consumption. Gasoline used for rice transplanters accounts for about 17% of the total fuel consumption each year. Tractors and rice transplanters emit 82% of all total pollutants. From 2011 to 2019, the total amount of air pollutant emissions decreased by 15%. That accounted for the reduction of rice cultivation fields in those periods. Rice transplanting operation accounts for 42% of total emissions. Then, harrowing, harvesting, tilling, leveling, and pest control operations generated 10%, 10%, 8%, 8%, and 7% of total emissions, respectively. The contribution of each air pollutant held 54% of CO, 39% of NOx, 5% of NMVOC, and 2% of TSP from the total emission inventory. The three major regions emitting air pollutants from mechanized agricultural practices were Jeollanam-do, Chungcheongnam-do, and Jeollabuk-do, which consume 55% of the total fuel usage in rice farming. The total amount of air pollutant emissions from rice cultivation practices in 2019 was calculated as 8448 tons in Korea.

**Keywords:** air pollutant emissions; rice cultivation; agricultural machinery; tier 1 methodology; geographic information system



## 1. Introduction

Rice (*Oryza sativa* L.) is a globally cultivated species and is the most cultivated crop in many Asian countries [1]. Rice (*Oryza sativa* L.) is the staple food for four billion people. The top three main rice producers in Asia are China, India, and Indonesia, while Korea is at position 15th in Asia [2,3]. Korea is one of the largest crop producing countries with $1643 \times 10^3$ ha of total arable area for crops and generated a total grain production of $4375 \times 10^3$ tons in 2015 [4,5]. Rice is a staple crop and it was cultivated in $730 \times 10^3$ ha in 2019 in Korea [4].

Traditionally, agriculture produces and consumes energy in one form or another, such as fossil fuel energy, electric energy, animal power, and human power [6]. The amount of energy used in agricultural production has increased intensively, because the traditional, low energy input farming is being replaced by modern high energy systems to produce more products efficiently [7]. Agricultural practices including tillage, planting,

fertilizer spreading, harvesting, etc., were done mostly by various types of machinery and have been recognized as a significant source of atmospheric particulate matter (PM) and gaseous pollutants which adversely affect human health and regional environment [8–18]. Agriculture is an industry that is directly or indirectly affected by climate change while emitting air pollutants through the use of various types of agricultural machinery in producing agricultural products [7].

Along with the mechanization of agricultural farming in Korea, more animal and human powers that used to be the main agricultural power sources in the past were substituted by agricultural machinery in the recent decades. Most of agricultural practices necessary to cultivate rice have been mechanized; the degree of mechanization of rice production was reported to be 98.6% in 2020 in Korea [19]. Air pollutants emitted from agricultural machinery have their origins mainly from the exhaustion of internal combustion engines. Fossil fuels used in internal combustion engines include diesel, gasoline, kerosene, heavy oil, etc. Among them, agricultural machinery uses mainly diesel and gasoline as their fuels in Korea. Given the growing importance of agricultural machinery, researchers started to estimate the emissions at the city, regional, or country level [20,21]. The agricultural machinery emissions of NOx, $PM_{10}$, VOCs, and CO were estimated at 16,209, 1348, 1933, and 7097 tons, respectively, in 2015, and 16,249, 1330, 1902, and 7038 tons, respectively, in 2018 in Korea [22]. The emissions of agricultural machinery were not negligible even though they were not evenly distributed in a whole year. They could be extremely large on preparing fields for seeding or planting and harvesting seasons. When agricultural machines are intensively used, their emissions could be comparable with on-road vehicles and play an important role on air quality [23].

The current agricultural machinery emission inventories were developed based on the machinery activity data (mileage, work output or fuel usage) and corresponding air pollutant emission factors [24–26]. The National Institute of Environmental Research (NIER) in Korea published a recommendation to estimate the amount of air pollutants from the use of various types of agricultural machinery [27]. A large number of data should be provided to calculate the yearly amount of eight air pollutants from agricultural machinery use. The emission factor, rated power, and load factor data are provided by the NIER handbook. However, other data including the number and working hours of each machinery type also are required for the calculation process. The European Environment Agency (EEA), on the other hand, adopts a somewhat different approach to calculate the amount of air pollutants from agricultural machinery [28]. Even with different methods developed, there is still a huge gap between current data and the real-world in-use activity. Firstly, the number of agricultural machines owned in a certain area cannot indicate the actual amount of machinery used. During the busy farming season, a large amount of agricultural machinery is rented and used, and some unused agricultural machinery may be included. Secondly, the working environment and utilization rate differ depending on the crop cultivation environment.

The objectives of the study were to analyze the emission characteristics of five main air pollutants by the EEA Tier 1 method for rice cultivation in Korea from 2011 to 2019 every two years. In addition, the spatial distribution of the amount of five pollutants was visualized by a geographic information system (GIS) on a country scale.

## 2. Materials and Methods

There are several important factors in the calculation of emission inventory.

### 2.1. Calculation of Air Pollutant Emissions

The Tier 1 method developed by the EEA was used to calculate air pollutants in the study. As shown in Equation (1), the calculation of emissions requires fuel consumption of various types of machinery for rice production and emission factors for each air pollutant emitted from agricultural engines.

$$E_{i,j,k} = \sum \{FC_{i,k} \times EF_j\} \tag{1}$$

where, $E_{i,j,k}$ is the amount of emission of pollutant from agricultural machinery of region; $FC_{i,k}$ is the amount of fuel consumption by agricultural machinery of region; $EF_j$ is emission factor of pollutant (kg/ton fuel); i is agricultural machinery type (i = 1, . . . , 4); j is type of air pollutant (j = 1, . . . , 5); and k is region (k = 1, . . . , 10).

### 2.2. Factors Related to Calculating Amount of Fuel Consumption

About four types of agricultural machines including tractors, power tillers, rice transplanters, and combine harvesters (harvester), are frequently used for rice cultivation in Korea. The power tiller is a two-wheel tractor, sometimes called a walking tractor, and performs various agricultural operations. The power tiller played an important role in the mechanization of Korean agriculture. The tractors are equipped with various implements such as plow, rotavator, and harrow, etc., and are used for tilling, harrowing, leveling, transportation, etc. Power tillers and harvesters are currently equipped with diesel engines and are mainly used for pesticide spreading and rice harvesting operations in the study, respectively. Rice transplanters are currently equipped with small gasoline engines. Data on the amount of fuel consumed per rice producing acreage could be obtained from the relevant literature published by the government [29]. Rice cultivation fuel consumption is classified according to the type of machinery and agricultural practices, and is listed as shown in Table 1 [21].

**Table 1.** Fuel consumption data for machinery and implements used in rice cultivation.

| Machinery | Practice | Fuel | OE [1] h/ha | FC [2] | |
|---|---|---|---|---|---|
| | | | | Liter/h | Liter/ha |
| Tractor | Soil Tilling | Diesel | 2.9 | 7.8 | 20.6 |
| | Soil Preparing | Diesel | 3.7 | 7.5 | 26.0 |
| | Soil Leveling | Diesel | 2.9 | 7.8 | 20.6 |
| | Others | Diesel | - | - | 40.2 |
| Power Tiller | Pesticide Spraying | Diesel | 9.9 | 2.0 | 19.8 |
| Transplanter | Transplanting | Gasoline | 3.5 | 3.3 | 8.4 |
| Harvester | Harvesting | Diesel | 2.6 | 10.7 | 27.8 |

[1] OE: Operation efficiency, [2] FC: Fuel consumption.

The cultivated area of rice by year was searched by the Statistics Korea. It is the rice cultivated area from 2011 to 2019 every two years. As shown in Table 2, rice cultivation areas of studied regions were prepared from 9 provinces and 1 total metropolitan city (TMC) including 8 metropolitan cities in Korea [4].

**Table 2.** Rice cultivation area of each region in Korea.

| Region | Rice Cultivation Area (ha) | | | | |
|---|---|---|---|---|---|
| | 2011 | 2013 | 2015 | 2017 | 2019 |
| CHB [1] | 44,504 | 42,893 | 39,786 | 33,069 | 33,247 |
| CHN [2] | 152,947 | 151,814 | 146,319 | 134,035 | 132,174 |
| GAW [3] | 35,955 | 33,968 | 32,300 | 29,710 | 28,640 |
| GYB [4] | 110,550 | 108,501 | 104,712 | 99,551 | 97,465 |
| GYG [5] | 91,727 | 88,949 | 82,071 | 78,484 | 76,642 |
| GYN [6] | 79,563 | 77,732 | 73,934 | 67,895 | 65,979 |
| JEB [7] | 130,696 | 126,799 | 121,765 | 118,340 | 112,146 |
| JEJ [8] | 430 | 302 | 128 | 113 | 45 |
| JEN [9] | 174,930 | 170,690 | 170,185 | 161,442 | 154,091 |
| TMC [10] | 32,521 | 30,977 | 28,144 | 30,072 | 29,384 |
| Total | 853,823 | 832,625 | 799,366 | 754,713 | 729,814 |

[1] CHB: Chungcheongbuk-do, [2] CHN: Chungcheongnam-do, [3] GAW: Gangwon-do, [4] GYB: Gyeongsangbuk-do, [5] GYG: Gyeonggi-do, [6] GYN: Gyeongsangnam-do, [7] JEB: Jeollabuk-do, [8] JEJ: Jeju-do, [9] JEN: Jeollanam-do, [10] TMC: Total of 8 metropolitan cities.

*2.3. Emissions Factors*

The emission factors of five air pollutants used in the study are listed in Table 3. They were obtained from the Air Pollutant Emission Inventory Guidebook from EMEP/EEA [20].

**Table 3.** Emission factors of diesel and gasoline fuel.

| Pollutant | Emission Factor (Unit: kg/ton Fuel) | |
|:---:|:---:|:---:|
| | **Diesel** | **Gasoline** |
| CO | 11.469 | 770.368 |
| NOx | 34.457 | 7.117 |
| TSP [1] | 1.913 | 0.157 |
| NMVOC [2] | 3.542 | 18.893 |
| $NH_3$ | 0.008 | 0.004 |

[1] TSP: Total suspended particles, [2] NMVOC: Non-Methane Volatile Organic Compounds.

*2.4. Spatial Allocation of Emission*

The total amount of emissions calculated in the study was visualized to understand the emission characteristics of each region. Emissions of five air pollutants have been assigned to each region using an open source GIS program (QGIS, version 3.10.11). The amount of emissions from the agricultural machinery used for rice cultivation is allocated to the vector layer according to the electronic geographic information.

## 3. Results and Discussion

*3.1. Amount of Fuel Consumption*

3.1.1. Amount of Fuel Consumption by Region

Table 4 shows diesel (D) and gasoline (G) fuel consumption of target regions. Gasoline is used only in rice transplanters, and diesel is used for the tractors, power tillers, and combine harvesters. Fuel consumption is in the order of Jeollanam-do, Gyeongsangnam-do, and Jeollabuk-do, accounting for 55% of the total fuel consumption for mechanized farming in Korean agriculture.

**Table 4.** Calculated amounts of diesel and gasoline usage.

| Region | Fuel Usage (Ton Fuel) | | | | | | | | | |
|:---:|:---:|:---:|:---:|:---:|:---:|:---:|:---:|:---:|:---:|:---:|
| | 2011 | | 2013 | | 2015 | | 2017 | | 2019 | |
| | D [11] | G [12] | D [11] | G [12] | D [11] | G [12] | D [11] | G [12] | D [11] | G [12] |
| CHB [1] | 6898 | 374 | 6648 | 360 | 6167 | 334 | 5436 | 295 | 5153 | 279 |
| CHN [2] | 23,707 | 1285 | 23,531 | 1275 | 22,679 | 1229 | 20,775 | 1126 | 20,487 | 1110 |
| GAW [3] | 5573 | 302 | 5265 | 285 | 5007 | 271 | 4605 | 250 | 4439 | 241 |
| GYB [4] | 17,135 | 929 | 16,818 | 911 | 16,230 | 880 | 15,430 | 836 | 15,107 | 819 |
| GYG [5] | 14,218 | 771 | 13,787 | 747 | 12,721 | 689 | 12,165 | 659 | 11,880 | 644 |
| GYN [6] | 12,332 | 668 | 12,042 | 653 | 11,460 | 621 | 10,524 | 570 | 10,227 | 554 |
| JEB [7] | 20,258 | 1098 | 19,654 | 1065 | 18,874 | 1023 | 18,343 | 994 | 17,383 | 942 |
| JEJ [8] | 67 | 4 | 47 | 3 | 20 | 1 | 18 | 1 | 7 | 0 |
| JEN [9] | 27,114 | 1469 | 26,457 | 1434 | 26,379 | 1430 | 25,024 | 1356 | 23,884 | 1294 |
| TMC [10] | 5041 | 273 | 4801 | 260 | 4362 | 236 | 4661 | 253 | 4554 | 247 |
| Total | 132,343 | 7172 | 129,057 | 6994 | 123,898 | 6714 | 116,980 | 6340 | 113,121 | 6130 |

[1] CHB: Chungcheongbuk-do, [2] CHN: Chungcheongnam-do, [3] GAW: Gangwon-do, [4] GYB: Gyeongsangbuk-do, [5] GYG: Gyeonggi-do, [6] GYN: Gyeongsangnam-do, [7] JEB: Jeollabuk-do, [8] JEJ: Jeju-do, [9] JEN: Jeollanam-do, [10] TMC: Total of 8 metropolitan cities, [11] D: Diesel, [12] G: Gasoline.

### 3.1.2. Amount of Fuel Consumption by Agricultural Machinery

Table 5 shows the fuel consumption by agricultural machinery from 2011 to 2019 every two years. Tractor, power tiller, rice transplanter, and combine harvester fuel consumption in 2011 were 64,549 tons, 16,906 tons, 23,736 tons, and 34,324 tons, respectively. Tractor, power tiller, rice transplanter, and combine harvester fuel consumption in 2019 decreased by 15% from 2011 to 55,173 tons, 14,450 tons, 20,289 tons, and 29,339 tons. Tractors account for 46% of total fuel consumption and 56% in the diesel fuel consumption. It can be seen that tractors are used more often than other agricultural machines. Gasoline used in rice transplanters accounts for 17% of total fuel consumption each year.

**Table 5.** Calculated amounts of fuel usage by agricultural machinery.

| Machinery | Implement | Fuel | Fuel Usage (Ton) | | | | |
|---|---|---|---|---|---|---|---|
| | | | 2011 | 2013 | 2015 | 2017 | 2019 |
| Tractor | Soil Tilling | D [1] | 17,589 | 17,152 | 16,466 | 15,547 | 15,034 |
| | Soil Preparing | D [1] | 22,199 | 21,648 | 20,783 | 19,623 | 18,975 |
| | Soil Leveling | D [1] | 17,589 | 17,152 | 16,466 | 15,547 | 15,034 |
| | Others | D [1] | 7172 | 6994 | 6714 | 6340 | 6130 |
| Power Tiller | Pesticide Spraying | D [1] | 16,906 | 16,486 | 15,827 | 14,943 | 14,450 |
| Transplanter | Transplanting | G [2] | 23,736 | 23,147 | 22,222 | 20,981 | 20,289 |
| Harvester | Harvesting | D [1] | 34,324 | 33,472 | 32,134 | 30,339 | 29,339 |
| | Total | | 139,515 | 136,051 | 130,613 | 123,320 | 119,252 |

[1] D: Diesel, [2] G: Gasoline.

### 3.2. Emissions of Air Pollutants by Agricultural Machinery

Figure 1 shows biennial air pollutant emissions from 2011 to 2019 by each agricultural machinery type. The total emissions from agricultural machinery is gradually decreasing. From 2011 to 2019, the total air pollutant emissions decrease by 15%. Tractors and rice transplanters are the two main sources of air pollutant emissions. They emit 82% of all total pollutants.

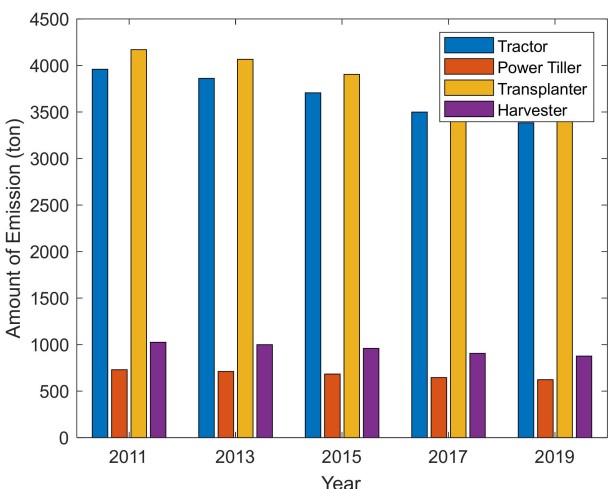

**Figure 1.** Biannual changes of total air pollutant emissions from agricultural machinery in rice cultivation.

Figure 2 shows the contribution rate of each agricultural operation to the total emissions of rice cultivation over the past nine years. Rice transplanting operation was the highest at 42%, harrowing and harvesting operations were 10% each, tilling and leveling operation was 8%, and pest control operation was 7%. The contribution rate for each air pollutant was CO 54%, NOx 39%, NMVOC 5%, and TSP 2%.

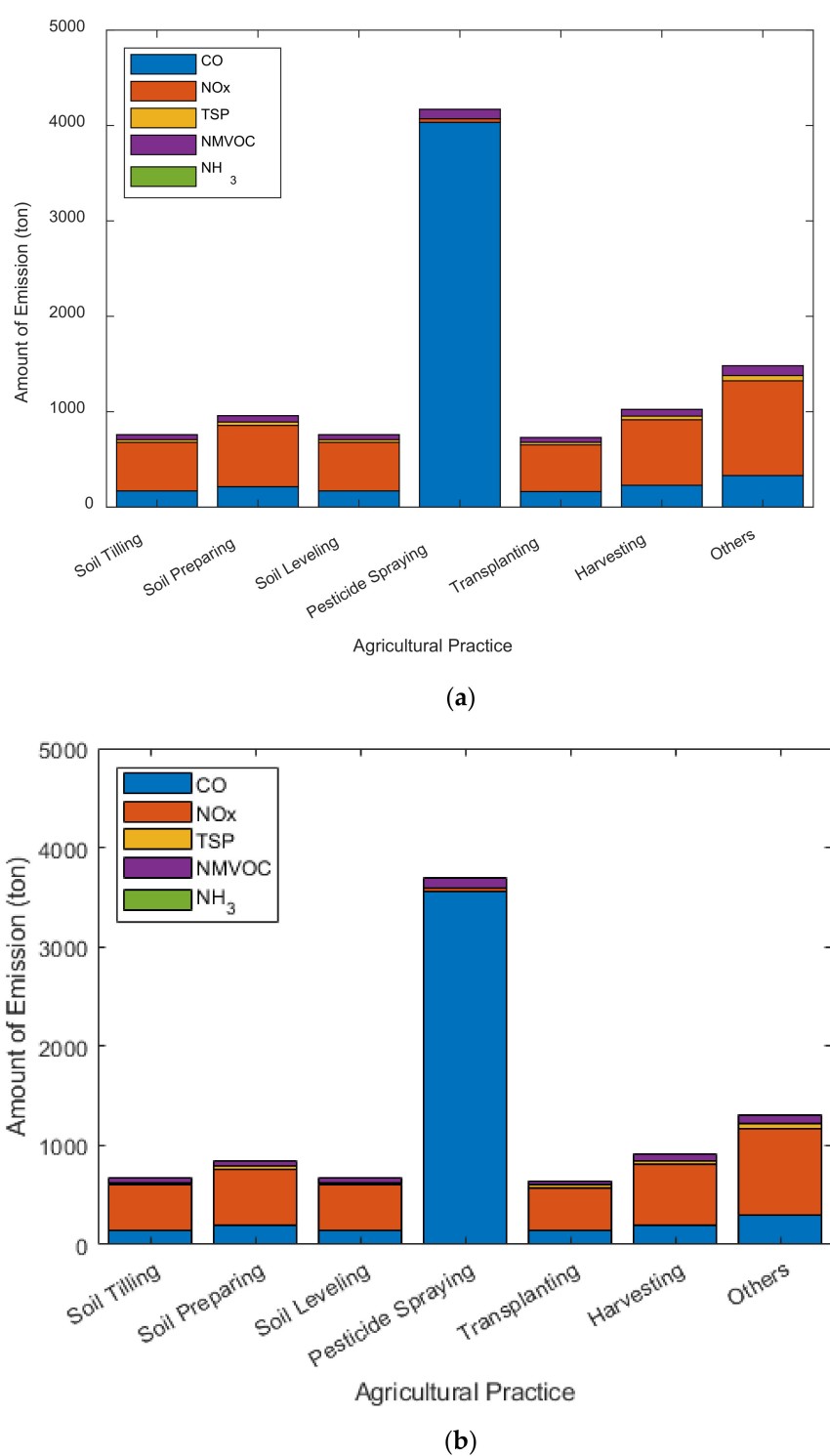

**Figure 2.** Calculated air pollutant emissions from various agricultural practices in rice cultivation: (**a**) 2011; (**b**) 2019.

### 3.3. Spatial and Temporal Distribution of Air Pollutant Emissions

Figure 3 shows biennial air pollutant emissions from 2011 to 2019. Annual emissions of air pollutants from rice cultivation by agricultural machinery are gradually decreasing. The total amount of air pollutants of CO, NOx, TSP, NMVOC, and $NH_3$ in 2019 were 4537 tons, 3306 tons, 182 tons, 421 tons, and 0.78 ton, respectively. The spatial distribution of air pollutants is shown in Figure 4. In general, the areas where rice cultivation emits air pollutants are concentrated in areas with large rice cultivated fields of about 398 kha.

Relatively low emissions are generally distributed in areas with low population densities and low rice production. In addition, economically developed large cities emit less air pollutants due to less agricultural activity.

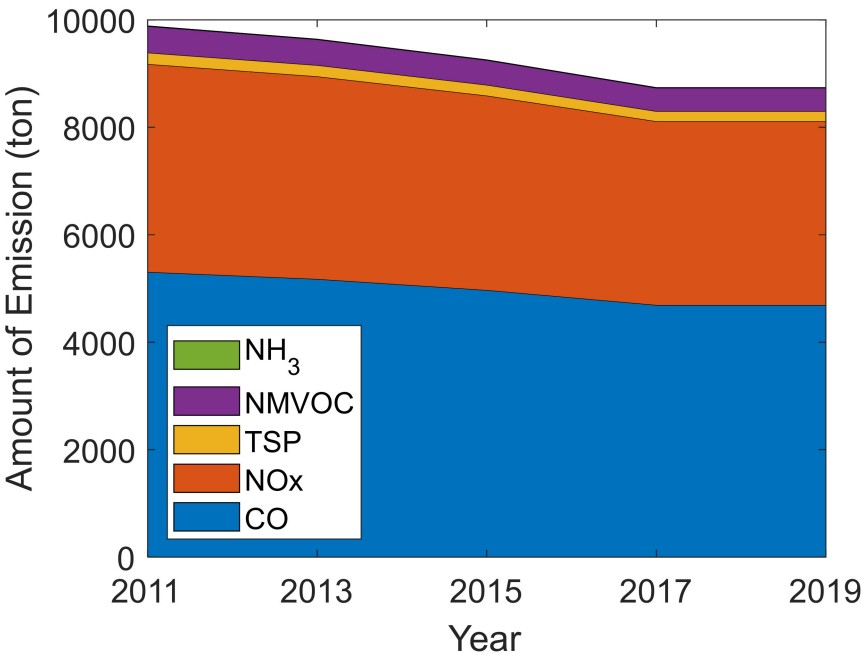

**Figure 3.** Biannual changes of air pollutant emissions in rice cultivation.

Tables 6 and 7 show the amount of each air pollutant emitted by each region in 2011 and 2019. The total amount of air pollutants was decreased by 1435 tons from 2011 to 2019. Based on the calculated data in 2019, the total emission of rice cultivated air pollutants in Korea is 8448 tons. The main emission area of air pollutants is the western part of the Korea, which has a huge plain for rice cultivation. Jeollanam-do, Chungcheongnam-do, and Jeollabuk-do account for 21%, 18%, and 15% of the total air pollutants. Gangwon-do and Chungcheongbuk-do, which are mountainous areas, have low emissions due to the influence of topography.

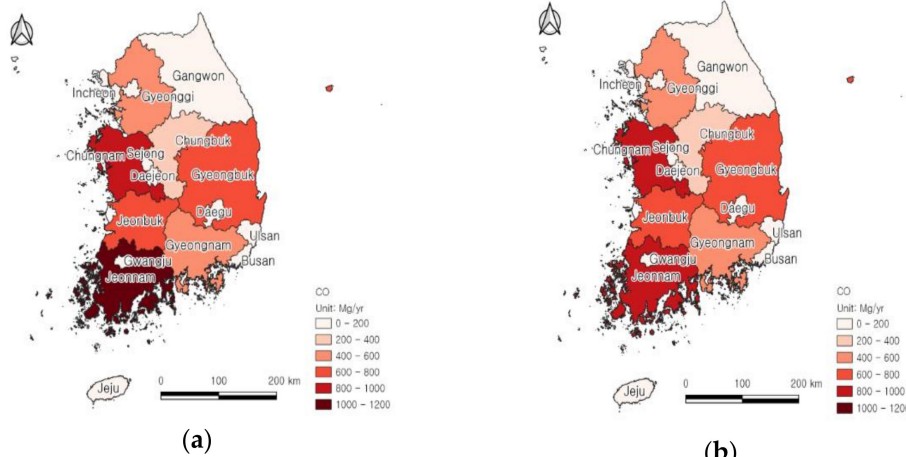

(**a**)         (**b**)

**Figure 4.** *Cont.*

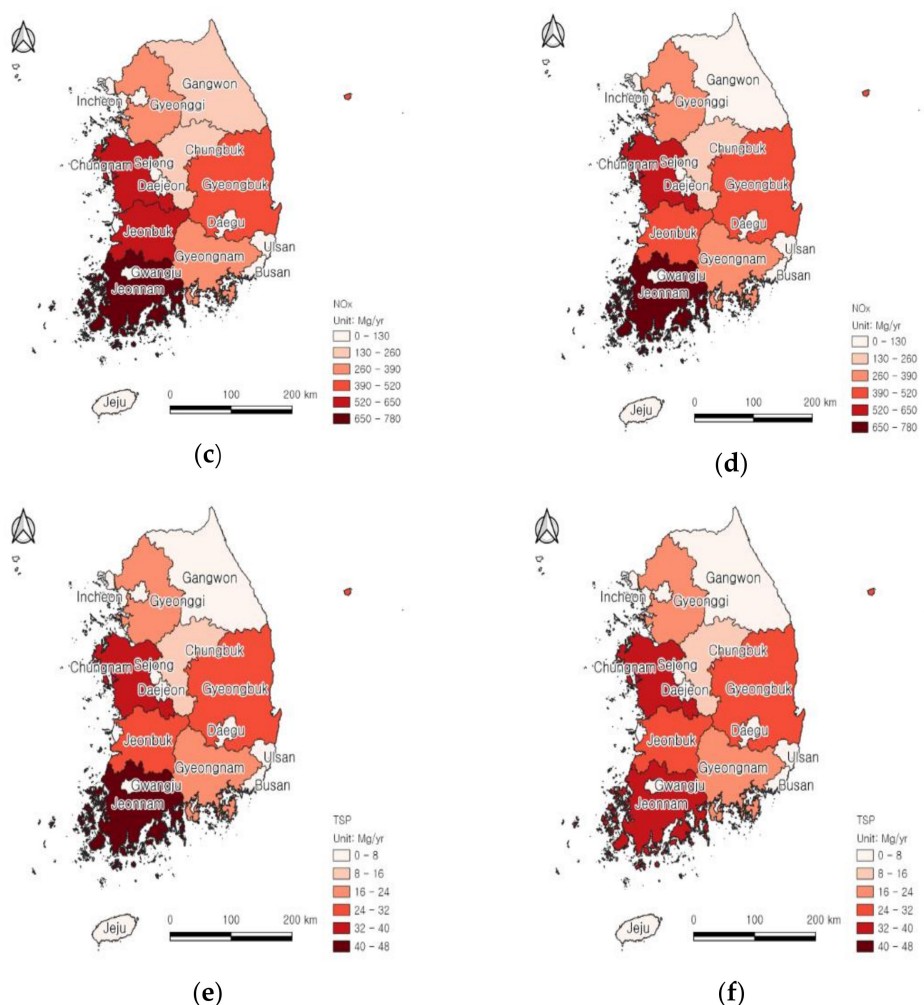

**Figure 4.** Spatial distribution of emissions of CO (**a**,**b**), NOx (**c**,**d**), and TSP (**e**,**f**) from Korean rice cultivation; (**a**,**c**,**e**) 2011; (**b**,**d**,**f**) 2019.

**Table 6.** Calculated air pollutant emissions due to rice cultivation by region in Korea (2011).

| Region | Emission (ton/yr) | | | | | |
|---|---|---|---|---|---|---|
| | **CO** | **NOx** | **TSP** | **NMVOC** | **NH₃ ($\times 10^2$)** | **Total** |
| CHB [1] | 277 | 202 | 11 | 25 | 5 | 515 |
| CHN [2] | 951 | 693 | 38 | 88 | 16 | 1770 |
| GAW [3] | 224 | 163 | 9 | 21 | 4 | 416 |
| GYB [4] | 687 | 501 | 28 | 64 | 12 | 1280 |
| GYG [5] | 570 | 416 | 23 | 53 | 10 | 1062 |
| GYN [6] | 495 | 360 | 20 | 46 | 8 | 921 |
| JEB [7] | 813 | 592 | 33 | 75 | 14 | 1513 |
| JEJ [8] | 2.673 | 1.948 | 0.108 | 0.248 | 0.046 | 4.977 |
| JEN [9] | 1088 | 792 | 44 | 101 | 19 | 2025 |
| TMC [10] | 202 | 147 | 8 | 19 | 3 | 376 |
| Total | 5308 | 3868 | 213 | 493 | 91 | 9883 |

[1] CHB: Chungcheongbuk-do, [2] CHN: Chungcheongnam-do, [3] GAW: Gangwon-do, [4] GYB: Gyeongsangbuk-do, [5] GYG: Gyeonggi-do, [6] GYN: Gyeongsangnam-do, [7] JEB: Jeollabuk-do, [8] JEJ: Jeju-do, [9] JEN: Jeollanam-do, [10] TMC: Total of 8 metropolitan cities.

**Table 7.** Calculated air pollutant emissions due to rice cultivation by region in Korea (2019).

| Region | Emission (ton/yr) | | | | | |
|---|---|---|---|---|---|---|
| | CO | NOx | TSP | NMVOC | NH$_3$(×10$^2$) | Total |
| CHB [1] | 207 | 151 | 8 | 19 | 4 | 385 |
| CHN [2] | 822 | 599 | 33 | 76 | 14 | 1530 |
| GAW [3] | 178 | 130 | 7 | 17 | 3 | 332 |
| GYB [4] | 606 | 442 | 24 | 56 | 10 | 1128 |
| GYG [5] | 476 | 347 | 19 | 44 | 8 | 887 |
| GYN [6] | 410 | 299 | 16 | 38 | 7 | 764 |
| JEB [7] | 697 | 508 | 28 | 65 | 12 | 1298 |
| JEJ [8] | 0.280 | 0.204 | 0.011 | 0.026 | 0.005 | 0.521 |
| JEN [9] | 958 | 698 | 39 | 89 | 16 | 1784 |
| TMC [10] | 183 | 133 | 7 | 17 | 3 | 340 |
| Total | 4537 | 3306 | 182 | 421 | 78 | 8448 |

[1] CHB: Chungcheongbuk-do, [2] CHN: Chungcheongnam-do, [3] GAW: Gangwon-do, [4] GYB: Gyeongsangbuk-do, [5] GYG: Gyeonggi-do, [6] GYN: Gyeongsangnam-do, [7] JEB: Jeollabuk-do, [8] JEJ: Jeju-do, [9] JEN: Jeollanam-do, [10] TMC: Total of 8 metropolitan cities.

The spatial distributions of major three air pollutants, CO, NOx, and TSP in 2011 and 2019 are shown in Figure 4. Major changes to the amount of CO and TSP are observed in the Jeollanam-do region. They indicate a huge reduction of rice transplanter in that region. NOx is decreased in Jeollabuk-do and Gangwon-do from 2011 to 2019, as shown in Figure 4. In general, the main rice producing regions are located in western parts, which emit the larger portion of air pollutants of Korea. Relatively low emissions are generally distributed in north eastern parts with mountainous terrains. In addition, economically developed large cities emit less air pollutants due to the lack of agricultural fields in Korea.

## 4. Conclusions

In this study, we calculated and analyzed five air pollutants emitted from four types of agricultural machinery from 2011 to 2019 in Korea. The total amounts of fuel consumed annually of tractors, power tillers, rice transplanters, and combine harvesters were calculated to estimate air pollutant emissions. Additionally, the amount of fuel consumed by nine regions was analyzed. The total yearly fuel consumption from 2011 to 2019 was gradually decreased from 139,515 tons to 119,252 tons of diesel and gasoline fuels. Tractors consumed 46% of the total fuel used for rice production. Rice transplanting operation generated the highest air pollutant emissions among various agricultural operations. It was also found that rice transplanters were the main source of CO emissions. The regions with larger air pollutants are Jeollanam-do, Chungcheongnam-do, and Jeollabuk-do. From 2011 to 2019, air pollution emissions emitting from rice cultivation decreased by 15%. It is presumed that the rice cultivation areas decreased, and so did air pollutant emissions.

The results of the analysis of air pollutant emissions in this study can improve the air quality management of local areas. Further study is needed to sophisticate the emission inventory in rice production using a more advanced methodology such as a Tier 2 method and more.

**Author Contributions:** Conceptualization, G.-G.H. and S.-M.K.; software, G.-G.H.; validation, G.-G.H., M.-H.K. and S.-M.K.; investigation, J.-H.J.; resources, G.-G.H.; writing—original draft preparation, G.-G.H.; writing—review and editing, M.-H.K., Y.-J.C. and S.-M.K.; visualization, J.-H.J.; supervision, S.-M.K.; project administration, S.-M.K.; funding acquisition, S.-M.K. All authors have read and agreed to the published version of the manuscript.

**Funding:** This work was carried out with the support of "Study on Particulate Matter Outbreak Source Characteristics during Agricultural Practice and Inventory Integration (Project No. PJ01428301)" Rural Development Administration, Korea.

**Institutional Review Board Statement:** Not applicable.

**Data Availability Statement:** Not applicable.

**Conflicts of Interest:** The funders had no role in the design of the study; in the collection, analyses, or interpretation of data; in the writing of the manuscript, or in the decision to publish the results.

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
