# Peer review of "Analysis of Air Pollutant Emissions for Mechanized Rice Cultivation in Korea"

_agriculture, doi:10.3390/agriculture11121208_

Round 1

Reviewer 1 Report

The authors have presented a study of pollutant emission from rice machinery in Korea from 2011 to 2019. The results show some regions have significant reduction due to change in farming circumstances

There are some minor changes that need to be addressed as outlined below

(1) Line 25: "...emissions was decrease by 15%." should be "...emissions decreases by 15%."
(2) Line 26: "Rice transplanting operation was in charge of 42% of.." ahaned to "Rice transplanting operation accounts for 42% of..."
(3) Line 41: "Korea is the one of the largest.." should be "Korea is one of the largest.."
(4) Line 132-136: This paragraph needs clarified. What does "was allocated visually" mean ?. Explain further on the spatial allocation of total emission and each of the machinery
(5) Line 163: "are decrease by 15%." Remove "are".
(6) Line 182: "large cultivated filed." should be "large cultivated field."
(7) Figure 4: colour scheme is hard to see. Use a better colour scheme
(8) Line 188: "by each region of 2011 and 2019" should be "by each region in 2011 and 2019"
(9) Line 205-206: "NOx is decreased and observed visually in Jeollabuk-do and Gangwon-do from 2011 to 2019" should be "NOx is decreased in Jeollabuk-do and Gangwon-do from 2011 to 2019 as shown in Figure 5"
(10) Page 224: "main regions" should be "main emission regions"
(11) Line 225: "was decreased by 15%." Remove "was"
(12) Line 228-229: This sentence needs qualification "It can contribute to establish effective policies to manage energy consumptions and to protect the health of farmers". You need an air quality model using the emission to do the assessment not only emission. 

Author Response

The authors have presented a study of pollutant emission from rice machinery in Korea from 2011 to 2019. The results show some regions have significant reduction due to change in farming circumstances

There are some minor changes that need to be addressed as outlined below

-> Thank you very much for your comments!

(1) Line 25: "...emissions was decrease by 15%." should be "...emissions decreases by 15%."

Revised as you mentioned.

(2) Line 26: "Rice transplanting operation was in charge of 42% of.." ahaned to "Rice transplanting operation accounts for 42% of..."

Revised as you mentioned.

(3) Line 41: "Korea is the one of the largest.." should be "Korea is one of the largest.."

Revised as you mentioned.

(4) Line 132-136: This paragraph needs clarified. What does "was allocated visually" mean ?. Explain further on the spatial allocation of total emission and each of the machinery

Revised clearly as you mentioned.

(5) Line 163: "are decrease by 15%." Remove "are".

Revised as you mentioned.

(6) Line 182: "large cultivated filed." should be "large cultivated field."

Revised as you mentioned.

(7) Figure 4: colour scheme is hard to see. Use a better colour scheme

Changed as you mentioned.

(8) Line 188: "by each region of 2011 and 2019" should be "by each region in 2011 and 2019"

Revised as you mentioned.

(9) Line 205-206: "NOx is decreased and observed visually in Jeollabuk-do and Gangwon-do from 2011 to 2019" should be "NOx is decreased in Jeollabuk-do and Gangwon-do from 2011 to 2019 as shown in Figure 5"

Revised as you mentioned.

(10) Page 224: "main regions" should be "main emission regions"

Revised as you mentioned.

(11) Line 225: "was decreased by 15%." Remove "was"

Revised as you mentioned.

(12) Line 228-229: This sentence needs qualification "It can contribute to establish effective policies to manage energy consumptions and to protect the health of farmers". You need an air quality model using the emission to do the assessment not only emission. 

Eliminated as you mentioned.

Reviewer 2 Report

Overall, the paper presents a well-integrated report on air pollution due to rice cultivation in Korea. The information in this paper can be of interest to researchers and policy makers in the country; besides, it can motivate research in other places and with other types of produce. From the technical point of view the paper is OK; however, there some issues that need to be corrected. They are listed below:

1.- The paper handles different type of units for pollutant emissions. They use tons, Mg and Ml. It is recommended to use tons throughout the paper.

2.- There are many grammar issues that need to be corrected. Some of them are highlighted in a marked file. A thorough revisions is suggested.

Author Response

Overall, the paper presents a well-integrated report on air pollution due to rice cultivation in Korea. The information in this paper can be of interest to researchers and policy makers in the country; besides, it can motivate research in other places and with other types of produce. From the technical point of view the paper is OK; however, there some issues that need to be corrected. They are listed below:

-> Thank you very much for your comments!

1.- The paper handles different type of units for pollutant emissions. They use tons, Mg and Ml. It is recommended to use tons throughout the paper.

Revised as you mentioned.

2.- There are many grammar issues that need to be corrected. Some of them are highlighted in a marked file. A thorough revisions is suggested.

Revised all as you mentioned on the manuscript.

Reviewer 3 Report

  • Introduction part: The introduction could, however, describe public policies for reducing emissions in the country, and in the agricultural sector in particular.
  • Line 57: The authors write that mechanization is gradually evolving, but it would be useful to understand the age and ecological level of the technology/equipment introduced. It would also be useful to understand the number of machines over the analyzed years, so that changes in fuel consumption and emissions can be understood in context. The most common brands can also be mentioned, as the foreign reader does not know the specifics of the country.
  • Line 94: The research methodology should be clarified.
  • Line 123: A detailed explanation of the reasons why 2011-2019 was chosen for the analysis would be necessary.
  • Line 162: A detailed explanation of the reasons for the reduction would be needed.
  • Line 191: How large plain is it?
  • Line 194: How large reduction in emissions?
  • Line 205: How large reduction of rice transplanters are?
  • Line 184/185 and Line 209/210: Almost the same statement was repeated in two places. An adjustment should be made.
  • Line 212: Figure 5 is difficult to see. The picture should be enlarged.
  • Conclusion part: It would be useful to see a future forecast for emission reduction, changes in agricultural land and fuel (perhaps even alternative) consumption. This is especially important at a time when efforts are being made to reduce emissions in all areas. It is possible that this correction could also be included in the "Results and discussion" section if the authors carry out additional research.
  • Line 227/228: Suggestions are needed on how to make improvements.
  • Line 229: Are data available on the health status of farmers at this stage?

Author Response

Comments and Suggestions for Authors

  • Introduction part: The introduction could, however, describe public policies for reducing emissions in the country, and in the agricultural sector in particular. 
  • -Korean government is starting emission reduction policy in agricultural machinery sector. This is intermediate results of the first national research project.
  • Line 57: The authors write that mechanization is gradually evolving, but it would be useful to understand the age and ecological level of the technology/equipment introduced. It would also be useful to understand the number of machines over the analyzed years, so that changes in fuel consumption and emissions can be understood in context. The most common brands can also be mentioned, as the foreign reader does not know the specifics of the country. 
  • -We don’t have detailed data in agricultural machinery. This research is using farming area and governmental data about duty-free fuel consumption for agricultural machinery only.
  • Line 94: The research methodology should be clarified. 
  • - This research is using Tier 1 method developed by the EEA using farming area and duty-free fuel consumption data for agricultural machinery only.
  • Line 123: A detailed explanation of the reasons why 2011-2019 was chosen for the analysis would be necessary. 
  • -We just want to study the trend of fuel consumption in rice production sector in the study.
  • Line 162: A detailed explanation of the reasons for the reduction would be needed.
  • Line 191: How large plain is it? 
  • -Revised as you mentioned.
  • Line 194: How large reduction in emissions?
  • -Revised as you mentioned.
  •  
  • Line 205: How large reduction of rice transplanters are? 
  • -We don’t need to know the number of reduction of rice transplanters in this study.
  • Line 184/185 and Line 209/210: Almost the same statement was repeated in two places. An adjustment should be made. 
  • -Revised as you mentioned.
  • Line 212: Figure 5 is difficult to see. The picture should be enlarged. 
  • -Changed a little bit because the manuscript could be too lengthy.
  • Conclusion part: It would be useful to see a future forecast for emission reduction, changes in agricultural land and fuel (perhaps even alternative) consumption. This is especially important at a time when efforts are being made to reduce emissions in all areas. It is possible that this correction could also be included in the "Results and discussion" section if the authors carry out additional research. 
  • -This study was carried out to establish emission inventory in rice production with Tier 1 method. Further works are undertaken to forecast future emission reduction.
  • Line 227/228: Suggestions are needed on how to make improvements. 
  • -Revised as you mentioned.
  • Line 229: Are data available on the health status of farmers at this stage?   
  • -> Thank you very much for your comments it will be very helpful for future researches!!
  •  
  • -Eliminated this paragraph for further study.

Round 2

Reviewer 3 Report

Dear authors,

Corrections are acceptable.